# The Betel Nut Intervention Trial (BENIT)—A Randomized Clinical Trial for Areca Nut and Betel Quid Cessation: Primary Outcomes

**DOI:** 10.3390/ijerph20166622

**Published:** 2023-08-21

**Authors:** Thaddeus A. Herzog, Lynne R. Wilkens, Grazyna Badowski, Ana Joy Pacilan Mendez, Adrian A. Franke, Pallav Pokhrel, Jade S. N. Chennaux, Lynnette F. Tenorio, Patrick P. Sotto, Crissy T. Kawamoto, Yvette C. Paulino

**Affiliations:** 1University of Hawaii Cancer Center, University of Hawaii at Manoa, 701 Ilalo Street, Honolulu, HI 96813, USAadrian@cc.hawaii.edu (A.A.F.);; 2University of Guam Cancer Research Center, University of Guam, Mangilao, GU 96923, USApaulinoy@triton.uog.edu (Y.C.P.)

**Keywords:** Betel Nut Intervention Trial, areca nut, betel quid, clinical trial, cessation, oral cancer

## Abstract

Background: Areca nut and betel quid (ANBQ) chewing is a widespread carcinogenic habit. The BENIT (ClinicalTrials—NCT02942745) is the first known randomized trial designed for ANBQ chewers. Methods: We compared the intensive behavioral treatment intervention condition (IC) with the control condition (CC) in the BENIT and included a 5-stage early stopping rule. We report the primary analysis at stage 3. English-literate adults in Guam and Saipan who self-identified as ANBQ chewers with tobacco were enrolled between August 2016 and August 2020. IC participants (*n* = 88) received five in-person sessions over 22 days and a brochure containing quitting advice. CC participants (*n* = 88) received only the brochure. Participants were assessed at baseline and on day 22 of follow-up. Self-reported chewing status at day 22 was determined by a composite of two survey items with disparate wording and response options for cross-verification. Results: Cessation rates were 38.6% (IC) and 9.1% (CC). Proportional hazards regression revealed a *p* = 0.0058, which met the Stage 3 criteria for significance, and an estimated reduction in ANBQ chewing for IC compared to the CC of 71% (95% CI: 41%–88%). Conclusions: Robust self-reported intervention effects at day 22 suggest that intensive cessation programs such as BENIT should be further developed and implemented on a larger scale.

## 1. Introduction

Areca nut (AN), the drupe fruit of the Areca palm (*Areca catechu*), is chewed by an estimated 600 million people worldwide [1], including the underserved Chamorro, Yapese, Palauan, Carolinian, and Chuukese ethnic minorities in Guam and Saipan of the Mariana Islands in the western Pacific [2]. During the years 2011 to 2015, approximately 11% of Guam’s population reported chewing on a regular basis [3], whereas the corresponding prevalence in Saipan was estimated at 24% in 2009 [4]. The majority of chewers in these islands prepare their AN wrapped in a pepper plant leaf (*Piper betle*) with tobacco, slaked lime (calcium hydroxide), and sometimes other spices [2] in what is referred to as a betel quid (BQ). According to the typology developed by Paulino et al. [2], these individuals are classified as Class 2 chewers. By comparison, those who chew AN alone (with the occasional addition of slaked lime) are referred to as Class 1 chewers.

In 2004, compelling evidence suggested strong associations between AN chewing and an increased risk of developing oral cancer (particularly oral squamous cell carcinoma), leading to the classification of AN chewing both with and without tobacco as carcinogenic to humans by the International Agency for Research on Cancer [5]. A subsequent meta-analysis further confirmed this association [6]. Survival at 5 years among oral cancer cases as low as 20% has been reported in some countries in the western Pacific region, which is in stark contrast to the average worldwide 5-year survival of over 50% [7].

Despite the global prominence of AN and BQ (ANBQ) and the devastating health burdens associated with their use, ANBQ remains an understudied research topic and an unappreciated global public health issue [8]. In addition, there is no comprehensive public health infrastructure for ANBQ prevention and control. This contrasts with substantial public health efforts to discourage tobacco consumption, particularly cigarettes. Likewise, research on ANBQ is in its infancy compared to the field of tobacco research [9].

One largely overlooked topic within ANBQ research is randomized trials of ANBQ cessation directed specifically at chewers who are ready to make a serious attempt to quit. However, there have been a few recent studies relevant to this issue. Lee et al. [10], for instance, published a qualitative investigation of ANBQ cessation in a sample of Taiwanese oral cancer patients. Tami-Maury et al. [11] conducted a qualitative assessment of potential ANBQ cessation strategies in a sample of Taiwanese dental clinics. Hung et al. [12] recently conducted a randomized trial of ANBQ cessation that employed anti-depressant medications. The results of that study were promising, though caution remains warranted due to the small sample sizes employed (*n* < 40 for treatment groups). Finally, Paulino et al. [13] published a report describing the protocol of the current randomized controlled trial to test the efficacy of an intensive behavioral ANBQ cessation program on the western Pacific islands of Guam and Saipan, aptly named The Betel Nut Intervention Trial (BENIT). BENIT is a registered clinical trial under the U.S. National Library of Medicine of the National Institutes of Health. This manuscript presents primary data consisting of self-reported cessation outcomes at the 22-day follow-up assessment. 

## 2. Materials and Methods

### 2.1. Study Design

The study design of BENIT was described in detail previously [13]. In brief, BENIT is a two-arm superiority randomized controlled trial (registered at clinicaltrials.gov, trial number NCT02942745). All data were collected in Guam (University of Guam) or Saipan (Community Clinic). Recruitment commenced in August 2016 and ended in January 2020. Data was collected between August 2016 and February 2020. Participants randomized to the intervention condition attended five in-person intensive behavioral intervention sessions administered over approximately 22 days. The intervention, developed using a smoking cessation intervention as an initial framework [14], was adapted and improved upon using findings from both surveys [3,15] and pilot intervention research [16] as part of a cumulative ANBQ research program in Guam. Thus, the BENIT intervention was developed in collaboration with local ANBQ chewers over a period of several years. In addition to receiving the intensive intervention, participants also received a booklet containing information about the negative health effects of chewing ANBQ and advice on how to quit chewing ANBQ. Participants randomized to the control condition received only the booklet. This study received IRB approval from the University of Guam (#16-04). The University of Guam IRB monitored the study for both study locations (Guam and Saipan) because Saipan does not have an IRB suitable for monitoring a randomized trial. All participants provided written informed consent and were compensated for their time. 

### 2.2. Target Population, Inclusion/Exclusion Criteria, and Recruitment

BENIT was conducted exclusively with Class 2 chewers residing in Guam and Saipan. Thus, all participants added tobacco to their BQ. This inclusion criterion was due to: (1) previous research that revealed a preponderance of Class 2 chewers in these locations [2], and (2) the fact that Class 2 chewers generally have a bigger health burden than Class 1 chewers, as the ANBQ with added slaked lime and tobacco is more addictive and more carcinogenic than AN alone [17].

Inclusion criteria were: (1) self-identified chewer of ANBQ with tobacco for at least one year with a minimum thrice weekly frequency; (2) at least 18 years of age; (3) willingness to attempt to quit chewing ANBQ during the intervention; (4) willingness to participate in five hour-long intervention sessions to take place over approximately 22 days; and (5) English literacy. Pregnant women were excluded from participating.

Recruitment was conducted using a variety of methods. Some participants were identified from previous ANBQ-related studies conducted under the National Cancer Institute-sponsored University of Guam/University of Hawaii Cancer Center Partnership to Advance Cancer Health Equity, while others were recruited through local community activities, health coalitions and associations, dental clinics, community health centers, village mayoral offices, radio announcements and interviews, religious organizations, on-campus university events, and print and social media advertising.

### 2.3. Sample Size and Early Termination Plan

The target sample size for BENIT was 324, which was based on a difference of 13 percentage points between intervention and control for ANBQ chewing cessation prevalence at 22 days with α = 0.05 (2-sided) and β = 0.20. However, an early termination plan was developed to accommodate the possibility of stronger-than-expected intervention effects. The specific plan employed the O’Brien-Fleming procedure for interim analysis to maintain the nominal type I error probability of 0.05. This procedure has been described previously [13]. BENIT employed a five-stage model for early termination that allowed for significance testing at pre-determined sample sizes. The first opportunity to test for significance (i.e., Stage 1) required 30 participants per study arm. To terminate the study at Stage 1, an absolute Z-score value of 4.5617 or greater was needed, with a corresponding *p*-value of 0.000005 or less. The Stage 2 analysis required 59 participants per study arm and a Z-score of 3.22564 or greater, with a corresponding *p*-value of 0.0012 or less for early termination. BENIT outcomes failed to reach these thresholds at Stages 1 and 2. The current article presents the findings at Stage 3, which required 88 participants per study arm. The current analyses pertain to the first 176 participants to complete 22-day follow-up assessments. 

### 2.4. Randomization

BENIT participants were randomized to each treatment condition within each island [13]. Randomization schedules were stratified by location (Guam and Saipan) and used blocked randomization [18], with random block sizes to avoid a large imbalance in size between the study groups during any particular timeframe. Separate randomization schedules were created for individuals and for groups in order to balance the randomization group size variable between intervention and control conditions. Condition assignments based on the randomization schedules were placed in opaque envelopes that were opened at the time of enrollment. The Biostatistics Core of the University of Guam/University of Hawaii Cancer Center Partnership designed the random allocation procedures. An independent group of research staff opened the envelopes at enrollment and informed participants of their random assignment to intervention or control. The Biostatistics Core personnel were blinded to the condition assignment as groups were labeled A and B for purposes of data analysis. Figure 1 displays the CONSORT diagram for participants included in the current analyses.

### 2.5. Control Condition Procedures

Participants in the control condition received a single booklet created specifically for BENIT and designed to encourage and facilitate ANBQ cessation. The booklet, entitled “Quitting Betel Nut”, included information about the health risks associated with ANBQ chewing as well as advice and strategies for quitting and maintaining ANBQ abstinence. Participants assigned to the control condition met with study staff three times: baseline, 22-day follow-up, and six-month follow-up. At baseline, participants were provided with the ANBQ cessation booklet, completed a baseline survey assessment, and provided a saliva sample. Participants also completed survey assessments and provided saliva samples at the 22-day and six-month follow-up sessions. Control condition participants received compensation for their time and specimen donation.

### 2.6. Intervention Condition Procedures

The BENIT intervention followed the general framework of intensive cognitive-behavioral therapy [19] and was influenced by an evidence-based smoking cessation program [14]. The intervention covered topics such as the negative health effects of ANBQ chewing, self-monitoring, triggers for chewing behaviors, substituting alternatives to chewing ANBQ, social support, and relapse prevention.

The intervention was comprised of five in-person sessions over a 22-day period. The BENIT program’s structure was such that participants were expected to attempt to quit chewing ANBQ at Session 3 (Day 15). Thus, Sessions 1 and 2 were intended to prepare participants for a quit attempt, whereas Sessions 3, 4, and 5 focused on relapse prevention. Saliva samples and survey assessments were collected at baseline (Session 1), at Session 5 (Day 22), and at 6-month follow-up (see Table 1). Intervention condition participants received the same compensation as control condition participants.

### 2.7. Baseline Assessments

Baseline survey assessments included questions on demographics, current chewing behavior, chewing history, ANBQ dependence measured by the Betel Quid Dependence Scale (BQDS) [20], and other variables. The BQDS is a validated 16-item scale that measures psychological and behavioral aspects of BQ dependence. BQDS scores range from zero to 16, with 16 representing the highest possible level of dependence. Baseline survey items were identical for participants in both study arms.

### 2.8. Follow-Up Assessments

The first follow-up assessment was administered at the final intervention session (Session 5). This assessment was scheduled for approximately 22 days after the first intervention session. Participants in this assessment provided data regarding their experience with the BENIT program and their chewing and quitting behaviors during the program. This follow-up assessment was identical for both intervention and control condition participants, except that control condition participants were not asked about the intervention program. 

The current analyses employed an outcome measure for the ANBQ that was a composite of two survey items. The first survey item read: “Which of the following do you consider yourself to be right now?” Response options were “Betel quid chewer” and “Ex-betel quid chewer”. The second survey item read: “Are you trying to quit chewing right now?” Response options were: “I already quit chewing”, “Yes, chewing but trying to quit”, and “No, chewing but not trying to quit”. For the current analyses, participants were coded as ex-chewers (quitters) if they responded “Ex-betel quid chewer” to the first survey item and “I already quit chewing” to the second survey item. We employed this composite variable to cross-verify self-reported cessation using two items with disparate wordings and response options. If the item responses provided a mixed (contradictory) response, the participant was categorized as a chewer. This composite outcome variable was employed for both intervention and control participants. 

### 2.9. Statistical Analysis

Participant characteristics were compared between randomization groups. A proportional hazards regression model of the time until cessation was fit using the randomization group as the independent variable, with and without adjustment for factors found to be out of balance between groups. The Wald test using a robust variance estimator accounting for the clustered (i.e., group) structure of the randomization was the test of the hypothesis. 

## 3. Results

### 3.1. Baseline Characteristics

Demographic and ANBQ chewing history characteristics are presented in Table 2 for the 176 participants included in the analyses. Participants were generally young, with a mean age of 28.5 years (SD 11.2), and 38.6% (*n* = 68) were female. The sample was comprised entirely of Pacific Islanders, with representation from several Pacific Islander ethnicities, including Carolinian, Chamorro, Chuukese, Palauan, Pohnpeian, and Yapese. The majority of participants resided in Guam. More than half of all participants graduated from high school, and roughly one-third had at least some college education. Participants chewed a mean of 10.3 times (SD 12.7) per day and had a mean BQDS score of 8.7 (SD 3.9). 

The following variables were compared between randomization groups: (a) age, (b) sex, (c) race/nationality, (d) study site, (e) participant group size, (f) number of ANBQ chews per day, (g) BQDS score, and (h) education level. The results revealed several substantial differences in baseline characteristics between the intervention and control conditions. Intervention condition participants were less likely to be female compared to control condition participants (30.7% versus 46.6%). Intervention condition participants also tended to be younger than control condition participants (mean years of 26.3 and 30.6 years, respectively), and participants in Guam were more likely to be randomized to the control condition as compared to participants in Saipan (see Table 2).

### 3.2. Cessation Outcomes

The current analyses included 88 participants in the intervention group and 88 participants in the control group. All participants included in these analyses provided outcome data at the 22-day follow-up. Although we aimed to schedule the initial follow-up assessments exactly 22 days following baseline, scheduling and logistical issues required that 22 days be an approximate timeframe for the follow-up assessments. The results indicated that 34 out of 88 (38.6%) participants in the intervention condition had self-reported ANBQ cessation (i.e., ex-chewers, according to the 22-day survey assessment), whereas 8 of 88 (9.1%) control condition participants self-reported cessation.

The proportional hazards regression model revealed a significant effect for the treatment group, with a Wald Chi-square test using a robust variance estimator (df = 1) = 7.6159, *p* = 0.0058. The Hazard Ratio for ANBQ chewing comparing the two randomization groups was 0.287 (95% CI: 0.123–0.592), indicating that the BENIT intervention led to a 71% reduction in ANBQ chewing as compared to the control condition. The Chi-square value of 7.6159 met the criteria for significance for early stopping at Stage 3: that Chi-square (df = 1) > 6.9365, or equivalently, that |Z| > 2.63372. (A Chi-square statistic with 1 degree of freedom is equivalent to a squared Z statistic). Thus, the null hypothesis that the cessation percentages for the intervention and control conditions are equal was rejected, and the BENIT ceased recruitment after the Stage 3 assessment.

An additional analysis of the treatment effect adjusted for the participant characteristics that were not in balance between treatment groups: sex (male or female), age (continuous), and location (Saipan or Guam). This analysis revealed a significant effect for the study condition: robust Wald Chi-square (df = 1) = 7.2017, *p* = 0.0073. The Hazard Ratio for ANBQ chewing between groups was 0.281 (95% CI: 0.111–0.710), indicating that the BENIT intervention led to a 72% reduction in ANBQ chewing as compared to the control condition after adjustment.

## 4. Discussion

This analysis of BENIT outcomes revealed strong self-reported intervention effects at the 22-day post-baseline assessment. Participant self-reports of 38.6% cessation for those in the intervention condition versus 9.1% cessation for those in the control condition met the Stage 3 criteria for early termination of the trial. This finding suggests that intensive cessation programs such as BENIT should be further developed and implemented in larger populations. 

Although the results of the current study are robust, several limitations should be noted. Bio-verification results for BENIT have been reported in a separate publication [21]. These results indicated that self-reported abstinence was consistent with salivary analyses at the aggregate level. That is, those reporting ANBQ abstinence revealed lower levels of ANBQ-associated alkaloids as compared to those reporting that they were current chewers of ANBQ. However, due to the developmental stage of our ANBQ biomarker research, we are not comfortable using these biomarkers to verify abstinence at the individual participant level at this time. We are continuing to develop and refine our research on ANBQ biomarkers so that future cessation trials can benefit from more established and systematic bio-verification metrics. Because bio-verification results cannot be employed reliably at the individual level, the current findings are based on participant self-reports. However, self-reported ANBQ abstinence was cross-verified using two different survey items. Further, participants were aware that their survey answers would be bio-verified with saliva samples collected at each assessment point, which likely encouraged accurate self-reports of chewing status. 

The results presented are ‘per protocol’ and did not include 29 participants (14 for control and 15 for intervention) who withdrew from the study prior to the 22-day assessment. Had these 29 cases been included (and counted as failures to quit), cessation rates would have been 33.0% for intervention (34 out of 103) compared to 7.8% (8 out of 102) for control. Randomization did not result in balanced arms for all potentially relevant variables. Specifically, several substantial differences in baseline characteristics were observed between intervention and control condition participants, particularly with regard to sex and age, though controlling for these variables did not affect the significance of the results. These differences were likely due to the allowance for groups to be randomized together. Further, the study involved many logistical and scheduling challenges (e.g., scheduling conflicts, postponements, and weather events). Recruitment was on a rolling basis and involved word-of-mouth effects (i.e., non-random effects). Thus, controlling for imbalances between intervention and control conditions needed to be accomplished statistically.

Recruiting participants for BENIT was challenging. There were several reasons for this. First, the populations of BQ chewers in Guam and Saipan are limited because the populations of these islands are not large. Second, the study was limited to those who wanted to quit (among other inclusion criteria). Third, participation in BENIT required a substantial commitment of time and energy on the part of participants (five in-person sessions). Fourth, our modest budget and research team limited the time we could spend on recruitment.

We acknowledge that we were not able to accomplish all of our original objectives as described in Paulino et al. [13]. This is because the research team was working with strictly limited resources. However, we believe that what we were able to accomplish is informative and hope that other researchers build on the BENIT experience. Further, and despite BENIT’s limitations, we believe that this study represents an important advancement for ANBQ cessation research.

Despite these limitations, it appears that the rationale for BENIT is supported and that a cessation program focused on ANBQ chewers who want to quit holds great potential for ANBQ chewers in Guam and Saipan, and possibly other countries in the western Pacific and other regions where ANBQ consumption is high. The results suggest that additional studies of intensive ANBQ cessation interventions are warranted. ANBQ cessation research should be expanded in many directions. Future developments should include: (1) continued refinement of biomarkers; (2) development of pharmaceutical adjuvants (similar to those used for tobacco cessation); and (3) additional treatment modalities such as electronic adjuvants for ANBQ cessation (for example, mobile applications).

Mehrtash et al. [9] recently suggested—and we agree—that progress in ANBQ research should build towards a framework similar to that which has been established for smoking cessation. Such a framework would entail elements such as clinical guidelines, evidence-based cessation and prevention interventions, validated biomarkers for ANBQ use, and an array of pharmaceutical products to aid cessation attempts. The success of the BENIT intervention makes a significant contribution towards that overarching goal.

## Figures and Tables

**Figure 1 ijerph-20-06622-f001:**
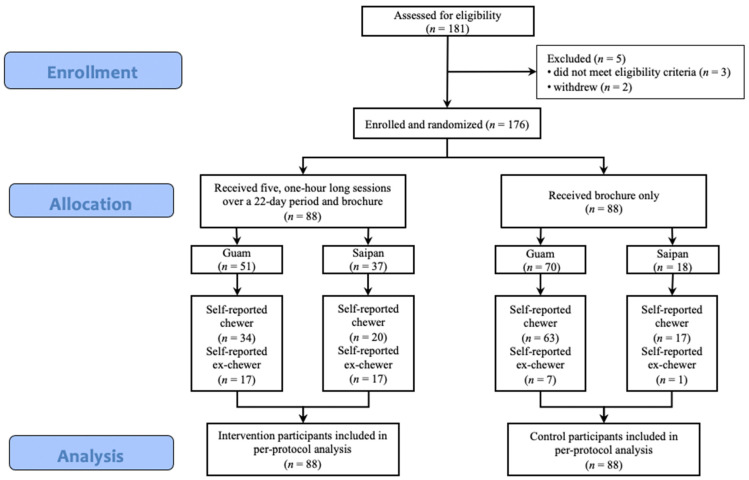
CONSORT flow diagram for the Betel Nut Intervention Trial.

**Table 1 ijerph-20-06622-t001:** Description of the intervention sessions for the Betel Nut Intervention Trial.

Session	Day	Description of Session
1	1	Introduction to the program. Obtain signed, informed consent. Health risks of areca nut and betel quid (ANBQ) chewing Introduce self-monitoring logs Introduce ‘trigger’ logs Discuss how to incrementally cut back on chewing prior to cessation Saliva collection and baseline survey
2	8	Discuss trigger logs and self-management Discuss lifestyle changes that encourage ANBQ cessation Preparing “excuses” for not chewing and the use of ANBQ substitutes Prepare for the upcoming Quit Day (morning of the next session)
3	15	QUIT DAY Encourage participants to be helpful and supportive to other participants regardless of chewing status Emphasize to participants that withdrawal symptoms generally last for 2 weeks before abating Reinforce coping techniques to prevent relapse Utilize social support to maintain chewing abstinence Emphasize alternative behaviors to chewing (e.g., physical activity)
4	18	Review and discuss quitting experiences, including coping strategies and different methods that participants used to manage triggers and avoid high-risk situations. Emphasize the immediate benefits of ANBQ cessation on health Review the negative health risks of ANBQ chewing, including the increased risk of oral cancer and other oral diseases Emphasize additional strategies for managing urges to chew ANBQ
5	22	Continue to support maintenance of chewing abstinence for those who quit Encourage participants who relapsed to chewing to attempt cessation again Teach approaches for managing thoughts and feelings that can result in relapse Reinforce lifestyle changes that are compatible with quitting ANBQ Review the use of “excuses” for not chewing and employing “fake chew” and other ANBQ substitutes Plan for the future, and discuss how to maintain ANBQ abstinence over time. Saliva collection and administer (22-day) survey assessment

**Table 2 ijerph-20-06622-t002:** Baseline participant characteristics.

Variable	Total (*n* = 176)	Intervention (*n* = 88)	Control (*n* = 88)
Gender, *n* female (% female)	68 (38.6)	27 (30.7)	41 (46.6)
Mean age, years	28.5 (11.2)	26.3 (9.8)	30.6 (12.1)
Ethnicity, *n* (%)			
Carolinian	27 (15.3)	14 (15.9)	13 (14.8)
Chamorro	37 (21)	22 (25)	15 (17)
Chuukese	50 (28.4)	23 (26.1)	27 (30.7)
Palauan	8 (4.5)	5 (5.7)	3 (3.4)
Pohnpeian	24 (13.6)	10 (11.4)	14 (15.9)
Yapese	16 (9.1)	7 (8)	9 (10.2)
Other Pacific Islander	14 (8)	7 (8)	7 (8)
Location, *n* (%)			
Guam	121 (68.8)	51 (58)	70 (79.5)
Saipan	55 (31.3)	37 (42)	18 (20.5)
Participants per Randomization Group	3.7 (3.9)	4.4 (4.4)	3.2 (3.2)
Participants per Treatment Group	NA	2.9 (3.2)	NA
Number of treatment sessions attended	NA	4.6 (0.6)	NA
Chews per day	10.3 (12.7)	8.3 (7.1)	12.3 (16.2)
Betel Quid Dependence Scale	8.7 (3.9)	9 (3.7)	8.5 (4.1)
Education, *n* (%)			
Less than high school graduation	77 (43.8)	38 (43.2)	39 (44.3)
High school graduate	35 (19.9)	22 (25)	13 (14.8)
Some college	27 (15.3)	10 (11.4)	17 (19.3)
Associate’s degree	29 (16.5)	16 (18.2)	13 (14.8)
Bachelor’s degree or higher	8 (4.5)	2 (2.3)	6 (6.8)

## Data Availability

Data will be shared pursuant to the data and specimen sharing policies and procedures of the Pacific Island Partnership for Cancer Health Equity (PIPCHE). Contact information: Hali R. Robinette at the University of Hawaii Cancer Center (hali@cc.hawaii.edu, Tel.: +808-564-5923).

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
