# Peer review of "The Betel Nut Intervention Trial (BENIT)—A Randomized Clinical Trial for Areca Nut and Betel Quid Cessation: Primary Outcomes"

_ijerph, 2023, doi:10.3390/ijerph20166622_

Round 1

Reviewer 1 Report

The authors have well-researched and under-researched areas. They have presented the entire manuscript very well as per recommended international guidelines.I have identified a few minor concerns that the authors must clarify or modify. There are a few suggestions to add into the manuscript to support their results.

1) there is a mention of class 1 and 2 chewers, please define them at the first occurrence for the benefit of novice readers.

2) Consent was informed but as it is a trial did the authors seek signatures to participate in the problem even though the intervention was non-pharmacological and no biospecimens were collected? 

3) was there any attrition even though they specified that analyses were done at stage 3. If possible briefly explain these stages for the readers. Instead of reading the reference

4) a brief table showing the baseline and end line (22 days) in the rate of quite and statistical; test of significance in this case chi square was done, but i presumed before after comparison of categorical outcomes need McNemar's test. Please check this and revise.

5) the essence of the regresssionwas to include all the baseline characteristics to examine of the observed difference between the two groups were impacted by any characteristics authors reported in the table 1. A full table of these results are desirable for journal paper on clinical trial.

Author Response

The authors have well-researched and under-researched areas. They have presented the entire manuscript very well as per recommended international guidelines. I have identified a few minor concerns that the authors must clarify or modify. There are a few suggestions to add into the manuscript to support their results.

  1. There is a mention of class 1 and 2 chewers, please define them at the first occurrence for the benefit of novice readers.

Response to 1. The distinction between class 1 and class 2 chewers was explained in the first paragraph of the Introduction (page 1, lines 33-38). We have reinforced the definition of class 2 chewers on page 2.

  1. Consent was informed but as it is a trial did the authors seek signatures to participate in the problem even though the intervention was non-pharmacological and no biospecimens were collected?

Response to 2. Written (signed) consent was collected for each participant. Please see page 2.

  1. Was there any attrition even though they specified that analyses were done at stage 3. If possible briefly explain these stages for the readers. Instead of reading the reference.

Response to 3. There was attrition. BENIT is a time-intensive program (five in-person sessions). Thus, it was inevitable that some participants would “drop out.” We reveal in the manuscript that there were 14 control and 15 intervention “drop-outs.” These participants were not included the 88 and 88 participants included in the stage 3 analyses. The stages are explained in section 2.3. We do not think that it would be feasible to explain fully the O’Brien-Fleming procedure in the current manuscript, but we appreciate the suggestion.

  1. a brief table showing the baseline and end line (22 days) in the rate of quite and statistical; test of significance in this case chi square was done, but i presumed before after comparison of categorical outcomes need McNemar's test. Please check this and revise.

Response to 4. This is an interesting suggestion. However, we believe that the data analysis approach that we employed is similar to, and consistent with, McNemar’s test. Both tests produce a chi-square test statistic. However, we believe the approach we employed is entirely appropriate.

  1. The essence of the regression was to include all the baseline characteristics to examine of the observed difference between the two groups were impacted by any characteristics authors reported in the table 1. A full table of these results are desirable for journal paper on clinical trial.

Response to 5. We appreciate these comments. We compared the baseline characteristics of the intervention and control participants for a lengthy list of variables. To account for variables for which significant baseline differences were detected, we provided an additional analyses of the outcomes while controlling for these variables. The difference in outcomes was negligible when controlling for these potential confounds. We believe that this practice is consistent with accepted practices for behavioral randomized trials such as BENIT.

Reviewer 2 Report

This article describes a novel RCT of a behavioural intervention for betel nut/quid addiction. The rationale is well-described. The article and development of the intervention is well-referenced to relevant literature.

Overall, the authors provide clear information about the trial design and methods, aligning (largely) with the CONSORT statement guidelines for reporting. However, there are some gaps and concerns:

First, the trial took four years to recruit just 176 participants. What accounts for such slow recruitment? Some consideration of this finding should be included. I am sure the authors can add insights here.

Second, there is no mention of efforts to co-design the intervention with end-users. Perhaps this has been done - if so, it should be noted.

Third, there is a lack of clarity about the pre-specified primary outcome and its analysis. Is it per-protocol or intention-to-treat? See my comments later. Was the self-reported data intended to be the primary outcome measure, or only the data verified by biochemical testing?

Fourth, the statistical analysis methods appear to be different from those indicated by Paulino in the paper referenced in reference 15, which seems to be an 'after the start of the trial' protocol paper. They should be the same. If not, why not?

Fifth, the inclusion of group-level intervention is commendable, because it affirms the cultural preferences of the target populations for group activities. However, I could see no discussion of how this was treated statistically (e.g. intra-class coefficient adjustment - is there any possibility of contamination of self-reported outcomes?) 

The a priori specification of stopping rules along the lines of the O'Brien-Fleming, although uncommon, is justified clearly and well-described.

The results are presented clearly. However, the per-protocol results are presented, not the intention-to-treat results, which should be the primary analysis reported here. I checked the aforementioned protocol paper (ref 15 Paulino et al) and could not discern this being pre-specified for the analysis. The fact that 14 and 15 people left the trial after randomisation is noted but the CONSORT flow chart includes them (there are 88 participants in each 'analysis' box, despite the boxes being labelled 'per protocol', so this is confusing).

The balance of baseline characteristics was not perfect and an adjusted analysis was done to take this into account, with little impact on the outcome. Was the adjusted analysis part of the initial analysis plan?

What biochemical tests are planned on participants' saliva? I had to look at the Paulino paper to find this information. I think it should be included here. Why are they 'pending'? Why not wait until the data on the biochemical verification of self-reported outcomes are available? It is unusual to publish without these data, (and not good practice if they were what was planned as the primary outcome analysis - again, more clarity is needed on this point).

Were the researchers and statisticians blinded to the allocation of the participants?

The discussion is thorough and includes a consideration of the strengths and weaknesses of the study and situates it in the broader context of this emerging field (betel dependence intervention). The authors include a succinct mention of some next steps in this field based on the experiences with this trial. Is there an appetite to conduct a larger trial, and if so, what would be changed or added? Following the tobacco cessation research playbook makes sense.
